# Daylight Saving Time and Acute Myocardial Infarction: A Meta-Analysis

**DOI:** 10.3390/jcm8030404

**Published:** 2019-03-23

**Authors:** Roberto Manfredini, Fabio Fabbian, Rosaria Cappadona, Alfredo De Giorgi, Francesca Bravi, Tiziano Carradori, Maria Elena Flacco, Lamberto Manzoli

**Affiliations:** 1Faculty of Medicine, Surgery and Prevention, University of Ferrara, via Ludovico Ariosto 35, 44121 Ferrara, Italy; roberto.manfredini@unife.it (R.M.); fabio.fabbian@unife.it (F.F.); rosaria.cappadona@unife.it (R.C.); 2Azienda Ospedaliero-Universitaria ‘S. Anna’, Via Aldo Moro 8, 44123 Ferrara, Italy; a.degiorgi@ospfe.it (A.D.G.); f.bravi@ospfe.it (F.B.); t.carradori@ospfe.it (T.C.); 3Regional Healthcare Agency of Abruzzo, via Attilio Monti 9, 65127 Pescara, Italy; elena.flacco@gmail.com; 4Center of Clinical Epidemiology, University of Ferrara, via Fossato di Mortara 64B, 44121 Ferrara, Italy

**Keywords:** daylight saving time, circadian rhythm, chronobiology, acute myocardial infarction, meta-analysis

## Abstract

Background: The available evidence on the effects of daylight saving time (DST) transitions on major cardiovascular diseases is limited and conflicting. We carried out the first meta-analysis aimed at evaluating the risk of acute myocardial infarction (AMI) following DST transitions. Methods: We searched cohort or case-control studies evaluating the incidence of AMI, among adults (≥18 y), during the weeks following spring and/or autumn DST shifts, versus control periods. The search was made in MedLine and Scopus, up to 31 December 2018, with no language restriction. A summary odds ratio of AMI was computed after: (1) spring, (2) autumn or (3) both transitions considered together. Meta-analyses were also stratified by gender and age. Data were combined using a generic inverse-variance approach. Results: Seven studies (>115,000 subjects) were included in the analyses. A significantly higher risk of AMI (Odds Ratio: 1.03; 95% CI: 1.01–1.06) was observed during the two weeks following spring or autumn DST transitions. However, although AMI risk increased significantly after the spring shift (OR: 1.05; 1.02–1.07), the incidence of AMI during the week after winter DST transition was comparable with control periods (OR 1.01; 0.98–1.04). No substantial differences were observed when the analyses were stratified by age or gender. Conclusion: The risk of AMI increases modestly but significantly after DST transitions, supporting the proposal of DST shifts discontinuation. Additional studies that fully adjust for potential confounders are required to confirm the present findings.

## 1. Introduction

Since decades, it has been shown that light-dark alternation influences the synchronization of circadian rhythms of most human systems [1]. Cellular processes show 24-h rhythms regulated by the biological clocks, which are ticking through a molecular clockwork, operated by a complex transcriptional/translational feed-back loop, hard-wired by core circadian genes and proteins [2,3]. The regulation of this complex machinery is extremely delicate, so that changing the time can disrupts body clocks and cause pathophysiological consequences far beyond the regulation of sleep [4].

The effects of time zone transitions (jet lag) are well known [5] and some meta-analyses showed chronobiologic variations in the occurrence of myocardial infarction, stroke, venous thromboembolism and aortic rupture or dissection [6,7,8,9]. Moreover, some studies reported a higher risk of cardiovascular diseases following daylight saving time (DST) transitions and it has been hypothesized that even a minor change in time schedule—such as the 1-h switch applied in about 60 countries worldwide—may cause a considerable stress for the body [4]. Because of these health concerns, in 2018 some northern Europe countries formally requested the European Parliament to abolish DST [10]. The Parliament solicited the European Commission and the scientific community to conduct an in-depth evaluation and a public survey. The public response strongly supported the discontinuation of bi-annual clock changes and the European Commission recently proposed their abolition [11]. Concerning the scientific evaluation of DST potential health effects, no meta-analysis has been published on the possible impact of DST transitions on acute myocardial infarction. The available evidence is fragmented and the results are complex to interpret by examining single studies. We thus performed a systematic review and meta-analysis in order to evaluate whether DST transitions may determine an increase in AMI risk.

## 2. Materials and Methods

### 2.1. Bibliographic Search, Study Selection Criteria and Quality Assessment

MedLine and Scopus databases were initially searched to identify studies that evaluated the incidence of acute myocardial infarction (AMI) during the week after (a) spring transition into DST and/or (b) autumn switch back from DST to standard time, versus the incidence of AMI during control periods (two weeks before DST for the spring transition; the second and third week after DST for the autumn transition). The bibliographic search was performed by two independent investigators (RM, LM) up to 31 December 2018, using the following search terms: (daylight saving time (Title/Abstract) OR DST (Title/Abstract)) OR (circadian rhythm (Title/Abstract) OR circadian misalignment (Title/Abstract)) OR (sleep (Title/Abstract) OR sleep deprivation (Title/Abstract)) OR (chronobiology (Title/Abstract)) AND (myocardial infarction (Title/Abstract) OR AMI (Title/Abstract)). While maintaining a common overall architecture, several alternative strings were used. The reference list of the reviews and retrieved articles were also searched for additional pertinent papers. There was no language restriction.

Studies were included if they: (a) had a cohort or case-control design; (b) included adult patients (≥18 years) with a diagnosis of AMI that was made in the first week after the shifts to and/or from DST and was documented through clinical chart, hospital discharge abstract databases or nationwide/local registries of pathologies; (c) provided enough data to compare AMI incidence before and after the time shifts.

Individual study quality was assessed using an adapted version of the Newcastle Ottawa Quality Assessment Scale [12]. We evaluated the comparability across groups at baseline (and examined whether the analyses were adjusted adequately for confounders), the appropriateness of the outcome assessment, the length of follow-up and the handling and reporting of missing data.

### 2.2. Data Analysis

Data were combined using a random-effect generic inverse variance approach [13], which enables the inclusion of diverse estimates of Relative Risk (i.e., odds ratio—OR—and hazard ratio—HR) into the same meta-analysis. From each paper, we extracted the adjusted estimates of AMI risk. When these were not available, we extracted the unadjusted estimates. If a paper reported the results of different multivariate models, the most stringently controlled estimates (those from the model adjusting for more factors) were extracted. If different models controlled for the same number of covariates, the model containing the most relevant covariates was used for the analysis. In case a study only reported separate ORs for each day of the week following DST transition, the summary weekly risk of AMI was computed from the separate estimates available using a fixed-effect meta-analysis of the individual study data [14].

The units of the meta-analysis were single comparisons of the rate of AMI in the first week following: (1) spring transition into DST; (2) autumn transition from DST; (3) both transitions considered together, versus the rate of AMI occurred during control weeks. Each of the above meta-analysis was also stratified by gender (computing summary ORs for males and females separately) and by age (<65 and ≥65 years). Thus, a total of 15 separate meta-analyses were performed.

Between-study heterogeneity was quantified using the I^2^ statistic [15]. The potential publication bias was assessed either graphically, using funnel plots (displaying ORs from individual studies versus their precision (1/standard error) and formally, through Egger’s regression asymmetry test [16]. However, formal tests for funnel plot asymmetry cannot be used when the total number of publications included for each outcome is <10, because the power is too low to distinguish chance from real asymmetry [13]. We were thus able to assess the publication bias only for one of the 15 meta-analyses.

All meta-analyses were performed using RevMan software, version 5.3 (Copenhagen: The Nordic Cochrane Centre, The Cochrane Collaboration, 2014). Stats Direct, version 3.1, was used to compute Egger’s test (StatsDirect Ltd., Altrincham, UK, 2017).

## 3. Results

### 3.1. Characteristics of the Included Studies and Methodological Quality

Among the 2633 retrieved papers, we identified seven studies (including >115,000 subjects) that met the selection criteria and were included in the analyses [17,18,19,20,21,22,23] (Figure 1). All were cohort studies, which retrospectively analysed data from national- or county-level disease registries [17,19,23], health insurance databases [21] or hospital discharge abstracts [18,20,22] (Table 1). All studies were published from 2008 onwards (five after 2012 [17,18,20,21,22]); five were based in a European country (Sweden, Finland, Germany and Croatia) [17,18,19,20,23] and two in USA [21,22]. Throughout Europe, DST started on the last Sunday of March and ended on the last Sunday of October during all study years [19]. In the studies from USA, spring and autumn shifts occurred on the second Sunday of March and on the first Sunday of November, respectively [22]. There was a wide variability in the number of patients enrolled, ranging from 935 to 42,000 and four studies had a sample size >10,000 [17,18,21,23]. Two studies did not provide data on the gender distribution of the participants [21,23]; in the remaining five studies, the proportion of males ranged between 59% and 73%.

The methodological characteristics of the included studies are summarized in Table 2. All studies adequately selected the cohort, assessed the exposure and outcome, described the losses to follow-up and the handling of missing data. The comparability of the participants was adequately addressed in four studies and only one paper reported some form of adjustment for potential confounders.

### 3.2. Overall Meta-Analysis: One Week after Both Spring and Autumn DST Transitions versus Control Weeks

In the analyses considering both shifts to and from DST, each study contributed with two datasets (one for the spring and one for the autumn transitions), thus a total of 14 datasets (with 116,675 participants) were included into the overall meta-analysis (Table 3 and Appendix A) [17,18,19,20,21,22,23]. As compared with control weeks, a significantly higher risk of AMI was observed during the week after both DST transitions (OR = 1.03; 95% CI: 1.01–1.06; *p* = 0.01).

In the analyses stratified by gender, three studies [17,19,20] reported separate ORs for males and females and one publication provided data for males only [18]. Therefore, a total of six datasets [17,19,20] (*n* = 10,382) and eight datasets [17,18,19,20] (*n* = 33,587) were included in the meta-analyses assessing AMI risk for females and males, respectively (Table 3; Appendix A). The risk of AMI did not significantly differ between post-transitional weeks and control periods for both women (OR = 1.02; 95% CI: 0.95–1.02) and men (OR = 1.02; 95% CI: 0.98–1.06). Similar results were achieved in the analyses stratified by age (Table 3; Appendix A): compared with control weeks, post-transitional AMI risk did not significantly increase among subjects aged <65 years (OR = 1.01; 95% CI: 0.97–1.05; *n* = 15,525) [17,19], as well as among the subjects aged ≥65 years (OR = 1.03; 95% CI: 0.97–1.08; *n* = 17,284) [17,19].

### 3.3. One-Week Post-Spring Shift to DST versus Control Weeks

A total of seven datasets [17,18,19,20,21,22,23] were included in the meta-analysis comparing the risk of AMI after spring DST shifts versus control periods. As shown in Table 3, a significantly higher risk was observed in the first week following the spring shift (OR = 1.05; 95% CI: 1.02–1.07; *p* < 0.001). In the analyses stratified by gender or age, no other significant difference emerged, with the only exception of the meta-analysis restricted to the subjects aged ≥65 years: OR = 1.07; 95% CI: 1.00–1.14 (Table 3; Appendix A).

### 3.4. One-Week Post Autumn Shift from DST versus Control Weeks

Seven datasets [17,18,19,20,21,22,23] were included in the meta-analyses comparing the risk of AMI in the seven days after the autumn shifts versus control weeks. No significant differences were observed, neither in the overall analysis, nor in the meta-analyses stratified by gender or age (all *p* > 0.05; Table 3; Appendix A).

### 3.5. Small Study Effects (Publication Bias)

In the single meta-analysis including ten or more studies (the one evaluating both spring and autumn post-transitional AMI risk), the funnel plot did not appear to be skewed and the corresponding Egger weighted regression tests did not identify asymmetry (*p* = 0.27—Appendix A).

## 4. Discussion

The main findings from this meta-analysis, based on seven studies including more than 100,000 participants, can be summarized as follows: (a) a significantly higher risk of AMI was observed in the weeks following spring and autumn DST transitions; (b) when spring and autumn DST shifts were considered separately, the risk of AMI increased significantly only in the week after the spring shift; (c) the increase of AMI risk, although significant, was relatively modest (≅3% overall; ≅5% after the spring transition); (d) in stratified analyses, which are likely affected by a lack of power, no substantial differences were observed by gender or age.

As the main potential explanation for the observed findings, DST transition may cause a disruption of the circadian rhythm, which in turn induces changes in sleep quantity and quality [24], together with a predominance of sympathetic activity, an increase in pro-inflammatory cytokine levels and a rise in heart rate and blood pressure [25,26]. Acting together, these factors have already been recognized as possible triggers of an increased cardiovascular risk following DST shifts [27,28]. Interestingly, although a misalignment between the internal circadian clock and the exogenous clocks should happen after both shifts [29], it has been suggested that the spring transition may be more disruptive on the circadian rhythm than the autumn transition [30,31], because turning the clocks forward induces one-hour shortening of the day and a likely sleep reduction [32,33]. This hypothesis seems to be confirmed by our findings of a significant increase in AMI risk after the spring shift, when the analyses were performed separately by season.

For most individuals, the alignment to the new time takes up to seven days [29,34], although a few studies suggested some degree of individual variation [29], essentially due to chronotype [30] and latitude [35]. Whether there is a difference across ages on the ability to cope with the circadian disruption induced by DST has been less widely assessed [20], but it has been hypothesized that the changes in sleep architecture due to aging (increased sleep fragmentation and latency [34]) may cause a prolongation of the time required to adjust the circadian misalignment [36]. Although we did find a borderline significant 7% increase in AMI risk among the elderly, our findings did not support entirely such hypothesis, which requires confirmation.

Despite the fact that about 1.6 billion people experience DST worldwide [30], with a mounting debate on its economic benefits [37], the available evidence on the health effect of the shifts in and out of DST is limited [11]: a few studies reported no effects of DST on the risk of stroke [36], manic episodes [38], suicide attempts [39] and spontaneous deliveries [40]. On the contrary, other publications reported an association between DTS shifts and several conditions including fatigue, headache, loss of attention and alertness, reduced motivation [34], traffic and workplace injures [33,41], missed medical appointments [42] and general mortality [11]. To the best of our knowledge, however, this is the first meta-analysis quantifying the potential effects of DST transitions on a severe, life-threatening condition such as AMI and the present estimates are the only currently available to quantify the overall cardiovascular burden following the shifts. Overall, our findings support the EU Parliament Research Service proposal of a DST shift discontinuation but additional quantitative estimates of the health burden of DST transition are needed to guide the decision process [10,11].

Several limitations should be considered in the interpretation of the results. First, most of our meta-analyses showed an intermediate-to-high level of heterogeneity. However, a certain degree of heterogeneity across studies may not be surprising given the large variation in terms of setting and baseline patients characteristics, which is typical of meta-analyses of observational studies [43]. Also, when the analyses were repeated adopting a fixed approach, none of the results substantially differed (except for CIs, which were typically tighter). Second, in all but one study [18], the exact timing of symptoms onset was not recorded, thus the time of hospital admission was consistently used as a proxy of AMI onset. Third, no study recorded data on several potential confounders of the association between DST and AMI, including sleep quality and quantity and chronotype [17,18,19]). Fourth, as previously noted, some of the age-and gender-specific estimates were based upon a limited number of studies and certainly require confirmation. Finally, although we made an extensive systematic search, we cannot exclude that additional data exist and were not considered.

## 5. Conclusions

This meta-analysis showed a modest but significant increase in the risk of AMI following DST transitions, that was particularly noticeable after the spring DST shift. Overall, these findings support the proposal of a DST transition discontinuation, although additional evidence is certainly needed to confirm the present results, identify high-risk subjects and quantify the relationship between DST transition and other severe diseases.

## Figures and Tables

**Figure 1 jcm-08-00404-f001:**
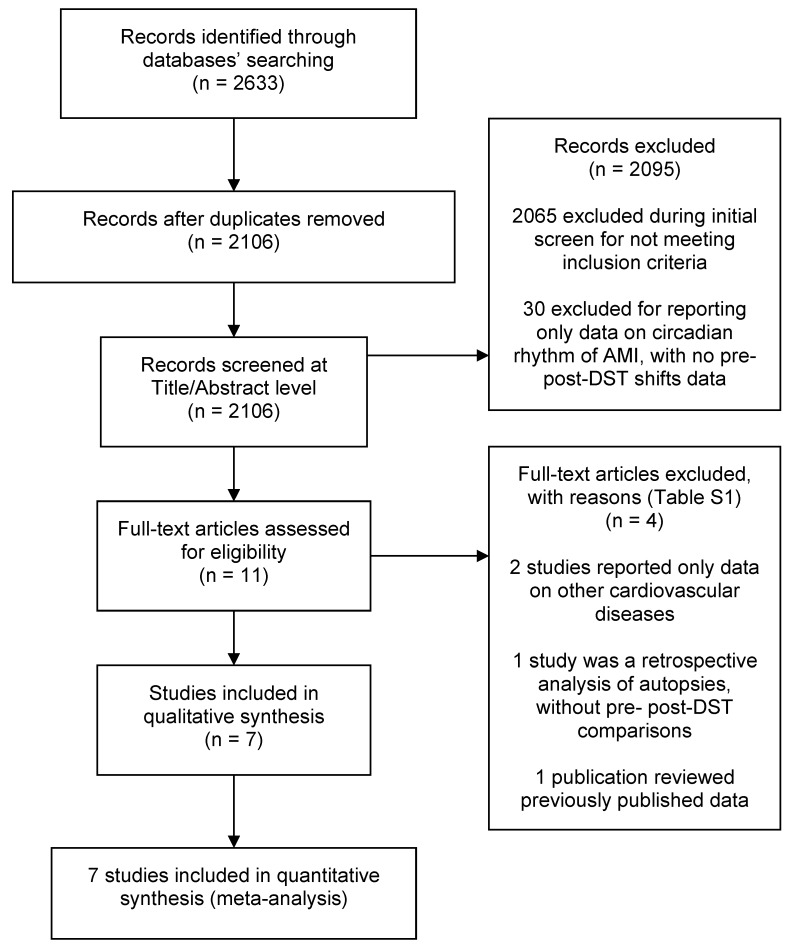
Flow of the included studies in each stage of the bibliographic search.

**Table 1 jcm-08-00404-t001:** Characteristics of the included studies.

First Author	Year	Country	Study Years	Data Source	Total Sample (Males)	Mean Age in Years (SD)
Janszky [23]	2008	Sweden	1987–2006	Swedish Registry of Acute Myocardial Infarction (no further detail provided)	>24,000 * (NR)	NR
Janszky [19]	2012	Sweden	1995–2007	Register of Information and Knowledge about Swedish Heart Intensive Care Admission (RIKS-HIA), including all patients with a diagnosis of AMI admitted to the Coronary care Units of 74 Swedish hospitals	>7300 * (>4650)	NR
Culic [20]	2013	Croatia	1990–1996	Hospital Discharge Abstracts of all patients with a primary diagnosis of AMI discharged from the Split University Hospital	2412 (1666)	64.1 (11.9)
Jiddou [22]	2013	USA	2006–2012	Hospital Discharge Abstracts of all patients with a primary diagnosis of AMI discharged from the Royal Oak and Troy Campuses of Beaumont Hospitals (Michigan)	935 (551)	70.0 (14.8)
Sandhu [21]	2014	USA	2010–2013	Blue Cross Blue Shield of Michigan Cardiovascular Consortium (BMC2) Database, including all Michigan residents with a diagnosis of AMI undergoing a PCI	42,060 (NR)	NR
Kirchberger [17]	2015	Germany	1985–2010	MONICA-KORA Myocardial Infarction Registry, including all residents in the County of Augsburg with a diagnosis of AMI, discharged from the Klinikum Augsburg Hospital (80%) or from minor County hospitals	25,499 (18,524)	62.6 (9.2)
Sipilä [18]	2015	Finland	2001–2009	Hospital Discharge Abstracts of all patients with a primary diagnosis of AMI (ICD-10 code 121x), discharged from one of the 22 Finnish hospitals with a coronary catheterization lab and treating emergency cardiac patients	14,459 (8748)	71.2 (12.8)

AMI = Acute Myocardial Infarction; PCI = Percutaneous coronary intervention; NR = Not reported. * When a study did not provide the total sample, we reported the overall number of patients with AMI among cases and controls.

**Table 2 jcm-08-00404-t002:** Methodological quality of the included studies according to the Newcastle-Ottawa Scale.

	Selection (Max Score 4)	Comparability (Max Score 2)	Outcome (Max Score 3)
Janszky [23]	4	0	3
Janszky [19]	4	0	3
Culic [20]	4	0	3
Jiddou [22]	4	1	3
Sandhu [21]	4	1	3
Kirchberger [17]	4	2	3
Sipilä [18]	4	1	3

**Table 3 jcm-08-00404-t003:** Risk of acute myocardial infarction (AMI) during the first week following daylight saving time (DST) transition versus control weeks *, overall and according to selected study characteristics. All meta-analyses are based upon a generic inverse-variance approach.

Variables	N. of Datasets ** (Sample) ^ϕ^	AMI Risk OR (95% CI)	*p*-Value	I^2^, %
1-week post-spring and autumn DST transitions vs. control weeks				
Overall [17,18,19,20,21,22,23]	14 (116,675)	1.03 (1.01–1.06)	0.01	67
Females only [17,19,20]	6 (10,382)	1.02 (0.95–1.09)	0.6	41
Males only [17,18,19,20]	8 (33,587)	1.02 (0.98–1.06)	0.3	25
Age < 65 years only [17,19]	4 (15,525)	1.01 (0.97–1.05)	0.6	0
Age ≥ 65 years only [17,19]	4 (17,284)	1.03 (0.97–1.08)	0.3	64
Spring shift—1-week post-transition to DST vs. control weeks				
Overall [17,18,19,20,21,22,23]	7	1.05 (1.02–1.07)	<0.001	24
Females only [17,19,20]	3	1.02 (0.88–1.18)	0.8	46
Males only [17,18,19,20]	4	1.06 (0.97–1.15)	0.2	49
Age < 65 years only [17,19]	2	1.01 (0.96–1.07)	0.9	68
Age ≥ 65 years only [17,19]	2	1.07 (1.00–1.14)	0.06	25
Autumn shift—1-week post-transition from DST vs. control weeks				
Overall	7	1.01 (0.98–1.04)	0.7	49
Females only [17,19,20]	3	0.99 (0.94–1.04)	0.8	0
Males only [17,18,19,20]	4	1.00 (0.97–1.04)	0.9	0
Age < 65 years only [17,19]	2	1.01 (0.96–1.06)	0.7	0
Age ≥ 65 years only [17,19]	2	0.99 (0.96–1.02)	0.5	0

OR = Odds Ratio; CI = Confidence Interval. * In all studies, control weeks are defined as the 2 weeks prior to DST and the 2 weeks following the 7 days after DST. ** In the analyses considering both spring and autumn shifts together, each study contributed with two datasets (one for the spring and one for the autumn transition) to the pooled estimates, thus the number of datasets is twice the number of included studies. ^ϕ^ The specific number of subjects included in each cohort (spring and autumn separately) was not available for all studies, thus only the sample size for the overall analyses was reported.

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
