# Peer review of "Daylight Saving Time and Acute Myocardial Infarction: A Meta-Analysis"

_jcm, 2019, doi:10.3390/jcm8030404_

Reviewer 1 Report

This review is meta-analysis evaluating the effect of daylight saving time transition on cardiovascular diseases. Authors have done significant work to put together the research about the topic. Although this topic is very interesting and important for readers, there are few issues to be address by authors to improve the quality of manuscript:

 1.      Abstract: The abstract should be revised for spelling/grammar mistakes.

 2.      Detailed review for spelling and grammatical errors is necessary; the text is cryptic in few places.

Author Response

I-1. The Referee wrote "This review is meta-analysis evaluating the effect of daylight saving time transition on cardiovascular diseases. Authors have done significant work to put together the research about the topic. Although this topic is very interesting and important for readers, there are few issues to be address by authors to improve the quality of manuscript: Abstract: The abstract should be revised for spelling/grammar mistakes".

 We agree and we accordingly revised the text of the Abstract. We are very sorry for the errors in the previous version.

 I-2. The Referee wrote "Detailed review for spelling and grammatical errors is necessary; the text is cryptic in few places".

 We agree and we apologize for the errors. We really did our best to improve the language and the readability of the manuscript, and we accordingly revised the text of all the sections of the manuscript. Please acknowledge that all changes have been highlighted in red.

Reviewer 2 Report

In this manuscript, Manfredini et al attempted to answer a very interesting question-- how daylight saving might impact the incidence of acute myocardial infarction, using meta-analysis. Meta-analysis has its own limitations which authors have discussed. The study is overall well designed and performed. Only one minor thing that the authors may need to mention in their analysis--- since the studies included in the analysis were from different countries, whether the DST occurs on the same day or on different days in different countries?

Author Response

II-1. The Referee wrote "In this manuscript, Manfredini et al. attempted to answer a very interesting question-- how daylight saving might impact the incidence of acute myocardial infarction, using meta-analysis. Meta-analysis has its own limitations which authors have discussed. The study is overall well designed and performed. Only one minor thing that the authors may need to mention in their analysis--- since the studies included in the analysis were from different countries, whether the DST occurs on the same day or on different days in different countries?".

 We entirely agree and accordingly added the following sentences in the first paragraph of the Results: "Throughout Europe, DST started on the last Sunday of March and ended on the last Sunday of October during all study years [19]. In the studies fromUSA, spring and autumn shifts occurred on the second Sunday of March and on the first Sunday of November, respectively [22]". We thank the Referee for the suggestion.